# Ethnic and Cultural Diversity amongst Yak Herding Communities in the Asian Highlands

**Srijana Joshi** [1,*] , **Lily Shrestha** [1] , **Neha Bisht** [1] , **Ning Wu** [2] , **Muhammad Ismail** [1] , **Tashi Dorji** [1] , **Gauri Dangol** [1] **and Ruijun Long** [1,3,*]

1 International Centre for Integrated Mountain Development (ICIMOD), G.P.O. Box 3226, Kathmandu 44700, Nepal; Lily.Shrestha@icimod.org (L.S.); neha.bisht16@gmail.com (N.B.); Muhammad.Ismail@icimod.org (M.I.); Tashi.Dorji@icimod.org (T.D.); gauri.dangol@icimod.org (G.D.)

2 Chengdu Institute of Biology, Chinese Academy of Sciences (CAS), No.9 Section 4, Renmin Nan Road, Chengdu 610041, China; wuning@cib.ac.cn

3 State Key Laboratory of Grassland and Agro-Ecosystems, International Centre for Tibetan Plateau Ecosystem Management, School of Life Sciences, Lanzhou University, Lanzhou 730000, Gansu, China

* Correspondence: Srijana.Joshi@icimod.org (S.J.); longrj@lzu.edu.cn (R.L.)

**Abstract:** Yak (*Bos grunniens* L.) herding plays an important role in the domestic economy throughout much of the Asian highlands. Yak represents a major mammal species of the rangelands found across the Asian highlands from Russia and Kyrgyzstan in the west to the Hengduan Mountains of China in the east. Yak also has great cultural significance to the people of the Asian highlands and is closely interlinked to the traditions, cultures, and rituals of the herding communities. However, increasing issues like poverty, environmental degradation, and climate change have changed the traditional practices of pastoralism, isolating and fragmenting herders and the pastures they have been using for many years. Local cultures of people rooted in the practice of yak herding are disappearing. Therefore, it is very important to document the socioeconomic and cultural aspects of yak herding. The broad aim of this paper was to provide a brief overview on the geographical distribution of yak in the Asian highlands and to provide in-depth information on yak-herding ethnic communities, the sociocultural aspect associated with yak herding, and challenges and emerging opportunities for yak herding in the Asian highlands. Altogether, 31 ethnic groups in 10 different countries of Asia and their cultures are documented herein. Yak was found to be utilized for many different household purposes, and to have cultural and religious aspects. Unfortunately, yak rearing and related traditions have been losing their charm in recent years due to modernization and several other environmental issues. Lastly, we suggest that there is an urgent need to take action to minimize the challenges faced by yak-herding mountain communities to conserve the traditional pastoral system and associated cultures of these ethnic communities.

**Keywords:** *Bos grunniens*; culture; ethnic diversity; herding; pastoralism; yak

## 1. Introduction

Among the 10% of the world's population living in Asia, about half resides in the Asian highlands [1]. Pastoralism is the prime source of livelihood for mountainous communities living in these regions. Yak (*Bos grunniens* L.) keeping forms one of the key part of extreme upland pastoralism, along with goat and sheep rearing, and is closely tied with the social and cultural life of the people, particularly in the vast rangelands of the Qinghai–Tibetan Plateau and other regions around the Himalayan mountain range [2,3]. Livestock contributes to about 20%–50% of the total income of pastoral and agro-pastoral communities in the Himalaya–Karakoram and Hindu–Kush region of the

Asian highlands [4]. In these mountain regions, different ethnic groups are involved in yak herding and follow a combined mountain agricultural practice including both crop cultivation and livestock husbandry with seasonal migrations between summer and winter pasture.

Fossil studies have suggested that the yak was found in northeastern Eurasia in the late tertiary period some 2.5 million years ago. These wild yaks migrated to the central area of the Himalayas, the Qinghai–Tibetan Plateau (QTP), after the formation of the mountains in the late Pleistocene epoch, which led to the formation of the alpine meadow habitat and a suitable climate for yak [5]. Wild yaks were caught and domesticated by ancient people in the Changthang area [5]. The domestication then spread to other highlands of the Himalayas. Although humans arrived in QTP very early, yak domestication and the significant increase in yak population are estimated to have come together (resulted from or contributed to) with the first and second human population expansions in the Asian highlands, respectively [6,7].

Yak herding is a major part of mountain livelihood where the terrain is not suitable for crop cultivation. Yak rearing plays a major role not only in the livelihood, but also in the cultural, religious, and social life of the people [8]. The yak can survive under harsh climatic and environmental conditions at high altitudes and provides a means of transport as well as a source of meat, dairy products, fiber, and hides. Thus, they are referred to as the "wealth" of local people, because products from yak provide income sources to the majority of people living in the highlands [8].

Apart from the economic significance, the domestic yak is also of great cultural and religious significance to the people living in the Asian highlands. They are closely linked to the traditional cultures and rituals of these herding communities. Along with other animals, yak is also indicated in the history, legends, and mythology, in both real and mythical form, of the Tibetan and its neighboring regions [9]. For example, the components of local medicine extracted from the yak are associated with one aspect of the near-mystical importance of the yak. Yak blood is regarded to have medicinal properties, and is taken from a juvenile's vein and fed to weak people in Nepal, especially in the Mustang area [10,11]. There are many documented studies about Tibetan people, but very limited information is available for other ethnic groups in the Asian highlands. Thus, the existing traditional knowledge of the biological and cultural systems of the yak-herding ethnic groups remains to be explored. Conservation, along with the sustainable use of biodiversity, is possible only when economies take into consideration this traditional indigenous knowledge and when they identify benefit-sharing as one of their goals. The broad aim of this paper was to provide evidence on the ethnic diversity and culture associated with yak husbandry in the Asian highlands. We obtained information on yak distribution, ethnic groups, culture, and religion by reviewing published and unpublished literature from peer-reviewed journal articles, books, edited book chapters, academic theses, livestock census data reports, and media articles.

The review focused on information about yak and yak-herding communities from the Asian highlands and the challenges faced by yak-herding communities. The Asian highlands comprises a mountainous region in Asia with a minimum altitude of 2400 m and a maximum of 8000 m [4].

The mountain systems located in the Asian highlands, starting from the north, are Sayan, Khangai, Altai, Tian Shan, Pamir, Hindu Kush Karakoram, Kunlun, Qilian, Tibetan Plateau, Hengduan, and Himalaya (Figure 1). This region is characterized by a harsh, cold climate of short summers and very long winters. The ethnic groups of this region rely on a traditional agro-pastoral mixed economy along with trade and barter practices.

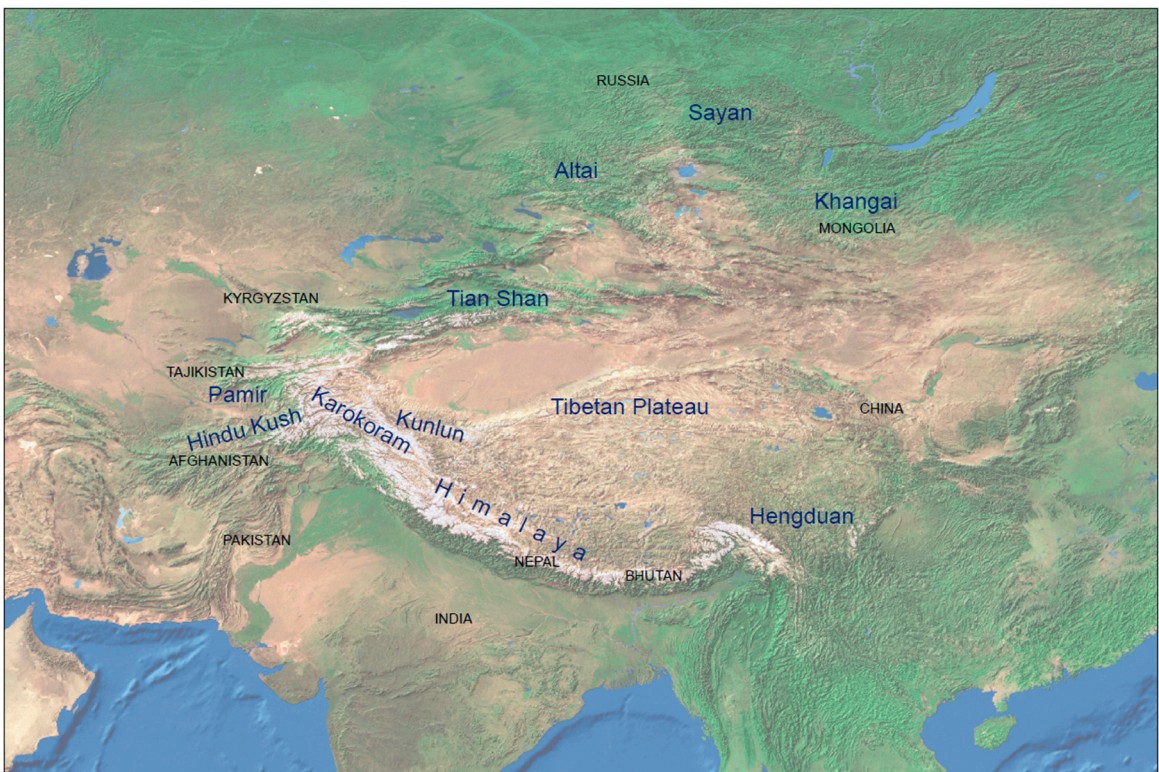

**Figure 1.** Map showing the mountain systems in the Asian highlands.

## 2. Distribution of Yak in the Asian Highlands

Yaks are distributed in 10 countries in the Asian highlands, namely Afghanistan, Bhutan, China, India, Kyrgyzstan, Mongolia, Nepal, Pakistan, Russia, and Tajikistan (Figure 2). In Afghanistan, yaks are limited to only Zebak and Wakhan of the Badakhshan province [2]. Likewise, yak production has been and continues to be the prime source of livelihood for people inhabiting the rugged landscape in Bhutan, extending from Haa in western Bhutan to Tashigang in the northeast [12]. China has the highest population of yak, with most of the yaks and their hybrids—about 94% of the population—concentrated in China [5]. Yaks are distributed in the Tibet Autonomous Region (TAR), Qinghai, Xinjiang, Sichuan, Gansu, and Yunnan provinces of China. States in north and northeast India, including Jammu and Kashmir, Sikkim, Himachal Pradesh, and Uttarakhand are home to several communities dependent on the yak [13]. In Central Asia, small population of yaks are kept in Kyrgyzstan, confined around the Chinese and Tajik borders [14]. Mongolia has the second highest population of yaks following China. In Mongolia, yaks are kept in 13 of the provinces, with 9 of these having 90% of the total yaks. About 70% of these yaks are located in the Hangai and Hovsgol mountains, 29% in the Mongolian Altai, and only 1% in the Gobi Altai and Hentii mountains [5].

Yak and yak–cattle hybrids are reared throughout northern Nepal in 29 districts, and many ethnic groups are involved in yak herding [15]. Yak herding in Pakistan is limited to the higher altitude areas of Gilgit Baltistan and Chitral [16]. Yaks are also reared in the northern Asian areas of Russia in Altai, West Sayan (Republics of Tuva), East Sayan (Republics of Buryatia) Mountains, and the Yakutia region [4]. Similarly, yak husbandry is practiced mostly in the eastern (Murghab), and some in the western (Ishkashim), Pamir region of the Gorno-Badakhshan region in Tajikistan [2].

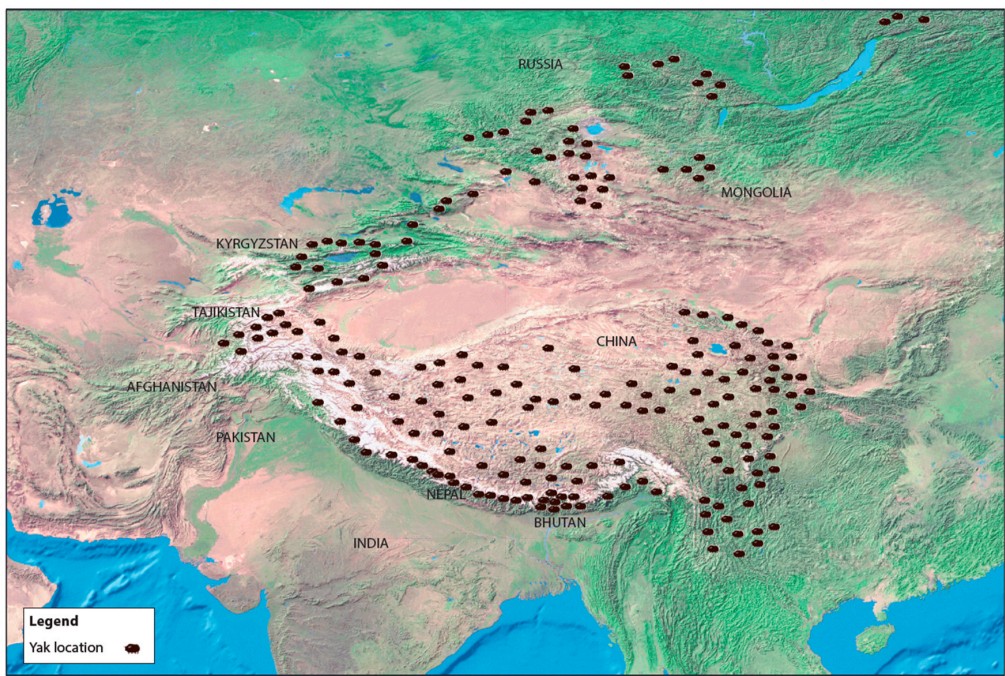

**Figure 2.** Distribution of yaks in the Asian highlands.

## 3. Ethnic and Cultural Diversity

Altogether, 31 major yak-herding ethnic communities were identified in 10 countries of the Asian highlands (Figure 3). Among them, Tibetans, Mongols, Kyrgyz, Wakhi, Sherpa, and Brokpa were the dominant yak-herding groups. Yak were used by all groups for similar purposes, such as pack and draft animals and for yak products derived from their milk, meat, hair, skin, and bones (Table 1). However, some ethnic communities were found to have some unique yak-derived foods and cultures.

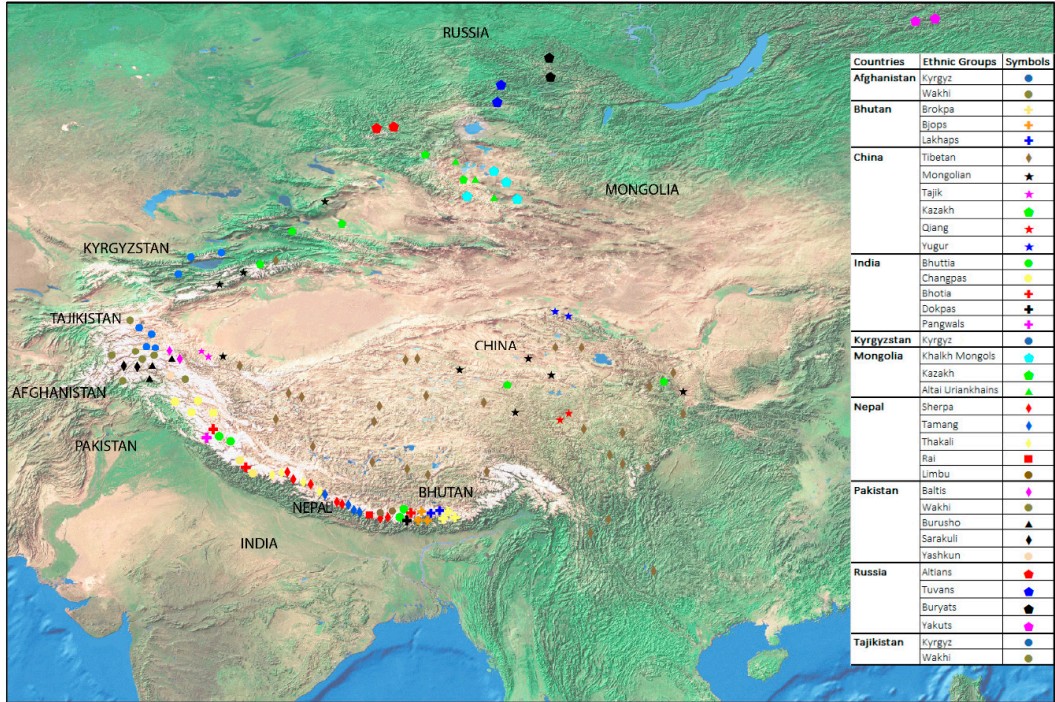

**Figure 3.** Major ethnic groups involved in yak herding in the Asian highlands.

### 3.1. Afghanistan

The total yak population in the Big Pamir, Wakhan district of Afghanistan was recorded as 1274 as of September, 2013 [17]. The Kyrgyz and the Wakhis are the two major yak-herding ethnic groups in Afghanistan, keeping yaks between 2000 m to 4500 m asl. The Kyrgyz inhabit higher altitudes and are a small population specialized in livestock rearing, whereas the dominant Wakhis are agro-pastoralists [18]. One of the main challenges to yak herders in Afghanistan has been the interruption of traditional migration and exchange processes and, thus, limited access to pastures in winter due to adverse political changes such as collectivization and central planning [18,19]. Other issues such as land limitation, policy issues, and climatic changes issues have also been causing consequences in both their crop and livestock production [19].

Kyrgyz

The Kyrgyz are nomads who move up to four times in a year, depending on the sun, wind, and pastures. They live in Yurt and their camps are located in both Great and Little Pamirs at altitudes between 4000 and 4500 m. Although they used to follow a long-distance nomadic transhumant migration cycle between the Pamirs and the lowlands of the mountains, their seasonal migration is now confined to Pamir regions only [18]. They breed yaks and also barter their yak products in exchange for basic amenities with people from Hunza, Pakistan [18].

Wakhi

Wakhi communities are distributed in remote mountainous areas of Pakistan, Afghanistan, Tajikistan, and China [2]. Wakhi people live in the Wakhan district of Afghanistan, which includes the Wakhan Corridor, the Big Pamir, and the Little Pamir. They are the Kyrgyz's closest neighbors. The Kyrgyz have cordial ties with the Wakhis, based mainly on bartering agricultural products with livestock products. The communities of Ismaili Shia Muslim Wakhis are located one level below the Kyrgyz at 2000–3000 m in the Wakhan Corridor [2]. They are agropastoralists. They migrate to high mountain pastures during summer and return to their permanent houses built on village lands with stone and mud-plastered walls.

### 3.2. Bhutan

According to Livestock Statistics 2017, published by the Department of Livestock of Bhutan, the present population of yak and crossbreeds in Bhutan is 50,334 head, owned by herders in 11 districts of Bhutan [20]. The ethnic groups involved in yak raising are known as the Brokpas and Dakpas in central and eastern Bhutan, Bjops in western Bhutan, and Lakhaps in the west–central region, keeping yaks from 2500 to 5000 m asl [21]. Yak-herding communities are scattered, marginalized, isolated with no access to roads and electricity, and have limited access to education and healthcare services, benefiting very little from the modernization of the economy [22]. The reliance of the yak-herding ethnic people on yak presents an uncertain future due to challenges such as shortage of pastures, low productivity of yaks resulting from inbreeding [23], changes in the climate, disease outbreaks, lack of resources for management of herds, and a shrinking market economy [24,25]. Climatic changes such as increased temperature and fluctuation in rainfall have been causing difficulty in herding and migration via reduction in grassland and thus yak production, which in turn has been causing negative consequences on herders' livelihoods [26]. Another challenge is the lack of nomadic centric government policies [27].

Brokpas

The Brokpas, often called Dakpas, are a minority ethnic group who mainly live in the villages of the Sakteng and Merak valleys in east Bhutan, located in the mountainous border region between eastern Bhutan at 3000 m asl under harsh climatic conditions. They speak Brokkat, a Tibetan–Burmese

language. They are semi-nomadic yak herders and also practice limited subsistence farming along with livestock rearing [12]. Yaks are reared in a traditional migratory system, taking the herds to higher elevation pastures up to 5000 m and staying there for few months during the summer, and travelling down to low altitudinal pastures to about 2500 m during the winter [12]. Brokpas have the largest share of the yak population compared to other ethnic group, and it is reported that they also sell 30% of their livestock products and barter the remaining 70% [28,29]. The Brokpa of Merak and Sakteng are popular for their fermented cheese in the eastern part of Bhutan and in lower altitudes [28].

Bjops

The Bjops are tribal groups of Bhutan who rely on yaks for their livelihood. Bjops were not originally yak herders, but they are said to have bought yaks from the Tibetans fleeing from the Chinese invasion [12]. Bjops mostly inhabit in Laya, Lunana, Lingzhi [shi], Soe, and Naro, with a few in Wangdue Phodrang [12,21]. Bjops communities have a migration pattern of moving to higher pastures during summer and lower pastures in winter [12]. They have a rich traditional culture, being a significant addition to the unique culture of Bhutan [22]. Bjops have a fascinating tradition of weaving a hundred-peg tent called a bja completely made out of yak hair [25]. Their myths about yak discovery and the splendid pastoral life of the herders are told to people through a special mask dance popularly known as yak Chham [29].

Lakhaps

The Lakhaps of Bhutan are an unengaged and unreached people in Wangdue Phodrang and Trongsa Districts in west–central Bhutan. They are part of the Tibetan people cluster within the Tibetan/Himalayan Peoples affinity bloc. This group of people is only found in Bhutan. Their primary language is Lakha, literally meaning "language of the mountain pass", and is spoken by the descendants of pastoral yak-herding communities. They follow the Buddhist religion and are semi-nomadic pastoralists.

*3.3. China*

There are about 13 million yaks in China, providing milk, meat, hair, and hide for Tibetan herders [30]. Yaks are primarily concentrated in the Qinghai–Tibetan Plateau of China, where they form the major source of livelihood and economy for the people living in the highlands. The majority of the yak-herding community in China is Tibetan, with other ethnic groups such as Qiang, Mongolian, Kazakh, and Yugur also keeping yaks in cold and semi-humid climates at altitudes ranging from 2000 to 4500 m asl. Yak in China are central to the religion, culture, and social life of people and communities who have been keeping yaks for over centuries [5]. Some of the issues impacting yak production systems include rangeland degradation, insufficient forage during winter, climatic uncertainty, mining impact and changing social values among younger generations [5,8,31]. Significant impacts on grass growth of rangeland due to delayed rainfall and shortened summers have caused decreases grassland productivity on the Tibetan plateau and, thus, decreases in livestock productivity. [32,33]. Increased uncertainty relating to climate-induced disasters such as snow-related disasters is also a growing risk for yak productivity in this region [32].

Tibetan

Tibetan people are a pastoral community with a majority population in the Tibetan Autonomous Region of China (90.48% of the provincial population), and also inhabit provinces such as Qinghai, Gansu, Yunnan, and Sichuan, with very few living in other provinces [34]. Livestock keeping is the basic industry of Tibetan people and yaks form a major part of the livestock husbandry adopted by Tibetan herders. Yaks are closely related to many aspects of the Tibetan Buddhist religion and their culture [5]. They follow traditional nomadic movement of herds between seasonal pastures with established houses or traditional Tibetan yak-hair tents, which they use as the base for their

migrations to distant pastures throughout the year [35]. The horns and skulls of yak hold special religious significance to Tibetan people, often engraved with mantras and placed in important places [8], and yak bodies are used as objects to drive away evil spirits. There is even have a special year to pay tribute to the Yak God in some Tibetan areas. Tibetans also play a traditional sport called yak racing in the traditional Wangguo festival in Lhasa, and many similar festivals are celebrated by herding communities each year in different part of the region [36]. The Yak Museum of Tibet, covering over 8000 square meters in Lhasa, was opened in May, 2014 with the primary mission of celebrating the "spirit of Yaks" [37]. It is the only museum in China, and in the world, that focuses on displaying the relationship of yak and nature, yak culture and spirit, and Tibetan history and culture.

### Mongolian

Mongolian people in China mostly live in Inner Mongolia, Qinghai, Gansu, and Xinjiang, with a population in China twice as high as that in Mongolia. Mongol nomads inhabit grazing lands in the Qaidam Basin and surrounding Kunlun Mountains in Qinghai Province and also in Inner Mongolia and Gansu [38]. They live in traditional Mongol tents or ger (yurts), and they keep livestock species such as yaks, goats, sheep, and horses [35].

### Tajik

The Tajik people inhabit in the Taxkorgan Tajik Autonomous County of western Xinjiang, where about 96% of the total Tajik people in China are located [39]. The primary source of income of the Tajik people is livestock husbandry of mountain goats, sheep, yaks, horses, donkeys, and camels following traditional nomadic style and also the practice of agriculture in the lower valleys. Each village in the Tajik community follows the tradition of using the pasture land collectively [39]. They live in yurts during the summer, and most of them have three houses, one in each of the summer, winter, and autumn pasturelands [39].

### Kazakh

The Kazakh people are also a Turkic ethnic minority of China, mostly living in the Xinjiang Uygur Autonomous Region and totaling about 1.1 million people in this region [40]. They practice nomadic herding of livestock and mainly keep sheep, goats, and yaks as their major livestock. Livestock herding is the basis of household income for Kazakh people in China [40]. The Kazakhs people also play traditional games such as buzkashi on yaks. The Kazakh people from China have started migrating to Kazakhstan in order to secure their pastoral life from the challenges of environmental degradation, political changes, and economic vulnerability [40].

### Qiang

The Qiang are a culturally and linguistically distinct ethnic minority of China [41]. They speak Qiangic languages and mostly follow the Qiang religion, with some following Tibetan Buddhism. Qiang people mainly inhabit the mountainous region of northwestern Sichuan on the eastern boundary of the Tibetan plateau along the Minjiang River and adjacent areas. Qiang are mainly yak herders and farmers and they practice yak grazing in high mountains in an agro-pastoral system. It is believed that ancient Qiang people were the first to tame and domesticate the wild yak, and they also practiced hybridization between yak and cattle some thousands years ago [5,42].

### Yugur

The Yugur are a Turkic or Mongolian ethnic minority living primarily in the Sunan Yugur Autonomous County in Gansu, China. They are Tibetan Buddhists speaking the western Yugur (Turkic) and eastern Yugur (Mongolic) languages. The Yugur people are predominantly involved in livestock

husbandry, which forms a major part of their livelihood. They live in moveable Tibetan-style square shaped tents sewn from rugs made of goat wool.

*3.4. India*

The total yak population in India according to the 19th livestock census in 2012 was 77,000, with the highest populations in the state of Jammu and Kashmir accounting for 71.08% of the total share. The recent 20th livestock census showed a decrease of 25% in yak populations from the previous census in India [43]. Bhuttia, Bhotia, Changpas, Dokpas, and Pangwals are the major yak-herding ethnic groups in the northern states of India, living at altitudes ranging from 3000–4500 m asl and following a semi migratory free-range system. These communities have been involved with yak rearing for a very long time and it has developed as part of their culture. However, they are facing a number of challenges to the sustainability of the yak sector such as extended seasons due to climate change, change in seasonal migration pattern of the yak, forest and rangeland degradation, decline in grazing area, shortage of feed and fodder, and modernization among younger generations [44,45].

Bhotia

The Bhotias, a group of Mongoloid origin, inhabit the high altitudes of Indian Central Himalaya at the Indo-Nepal and Indo-Tibetan borders. The Bhotias are traders and pastoralists, and until 1962, trade between India and the Tibet Autonomous Region of China was their primary occupation, with a change to agro-pastoralism in recent decades. However, they still practice transhumant herding. Wool industry is another element of the Bhotia economy, along with trade. They have specialized skills in processing the fur from domestic animals into woolen materials [46].

Bhuttia

The Bhuttias are the major yak-herding community in the northern Lachung and Lachen valleys of the northern districts of Sikkim [47]. They are Buddhist by religion, speak the Tibetan language, and practice semi-nomadic pastoralism where they take their yaks to higher pastures during summer and descend to lower pastures for winter [48]. Yak horns are considered holy and are also used for decorative purposes. The tail also has religious value. They are washed properly and tied with a rope tightly in a wooden handle to make Chamar(yak tail fan) used for deity worship [48]. The tails are also used as a fly whisker in some areas of India. Yak skins are used for decorative purposes and also to prepare hide, tents to resist cold, and mura (stool) [48].

Changpas

The Changpas (also called Champa, Fangpa, or Phalpa) inhabit the northern plain of Ladakh and Jammu and Kashmir in Changthang region [49]. They are divided into two groups—the sedentary Fangpa and the nomadic Phalpa. They follow a primitive form of Tibetan Buddhism and speak a Tibetan dialect, Changkyet/Chanskat. The Changpas have a pastoral life with yaks, dzo, and sheep being their chief economic sources. They migrate with the whole group of their domesticated herds between the lowlands and highlands according to the favorable seasons for their animals, and spend their whole lives in a large, portable yak hair tent called a rebos [49]. For many of them, rearing animals and selling their milk and meat products is their only means of livelihood. Changpas not only consume the products from their livestock but also barter them for grains and other utilities with the settled population [49].

Dokpas

The Muguthang area of Lhonak Valley, Lasher Valley, and the Tsho-Lhamu Plateau has been home to the nomadic pastoral community known as the Dokpas for centuries. This area has the last 23 families of the Dokpa community who breed pure Tibetan stock of yak [50]. Dokpas have

remained socioeconomically marginalized as they have continued to practice herding of yak, sheep, and pashmina-type goats.

Pangwals

The Pangwals live in the Pangi of Himachal Pradesh. Agriculture and livestock herding is an income source of all Pangwal communities. They are mainly Hindu with a small population of Buddhists in the higher villages called Bhotis. Both the native tribes of Pangi, Pangwals and Bhotis, are hardworking and robust people who have long preserved their unique culture, represented in folk songs, music, and tribal dances [51]. Yaks are also used as a tourist attraction in Indian Himalaya. Yak skiing is a special sport using yak that is practiced in the Indian hill resort of Manali, Himachal Pradesh as a tourist attraction [52].

### 3.5. Kyrgyzstan

Kyrgyzstan, located in the northeast of Central Asia, has 94% of its area covered by mountains with elevations stretching from 840 m in its capital to over 7000 m in the Tian Shan and Pamir-Alai ranges, a geography that has greatly influenced the semi-nomadic pastoralism system of Kyrgyz [53]. Livestock keeping is the only source of livelihood for a small population living in these regions. About 20,000 yaks are kept in Kyrgyzstan at altitudes of around 3500–4500 masl [5,54]. However, the natural conditions and resources available there can support 200–250 thousand yak without causing any harm to the natural resources and other animals [55]. The major challenges for Kyrgyz herders have been the shift to a free market economy from a planned economy resulting in a reduction in yak population and production, thus causing adverse economic impacts among the herders and inbreeding due to lack of yak bull exchange between the herds [53,54].

Kyrgyz

The cultures and traditions of Kyrgyz herding communities are still practiced in their purest form in the Issyk-Kul and Naryn provinces of eastern Kyrgyzstan. These areas mostly have mountain pastures and livestock herding is the prime source of livelihood [56]. The Kyrgyz people take their herds to different pastures for grazing according to the different seasons [18]. Historically, Kyrgyz people were pure nomadic herders and travelled from one pasture to another in a group [57]. However, they have adopted various coping strategies in their transhumant livestock herding after the end of state support for semi-nomadic herding such as changing migration pattern, remaining in large herding groups, migration as an extended family unit, or partnership with friends and neighbors, diversifying income-generating activities [56].

### 3.6. Mongolia

Archeological evidences has shown that Mongolians have been breeding yaks since 2500 years ago [58], and today, about 75% of Mongolia's population is involved in livestock husbandry [59]. The total population of yaks and yak hybrids in Mongolia is the second highest after China [5], with 676,300 in 2000 [5] and 507,600 in 2004 [58]. The major ethnic groups involved in yak herding in Mongolia are Khalkh Mongols, Kazakhs, and Altai Urainkhans, who keep yaks above the tree line at 2000 m in the north to 3000 m in the Mongolian Altai [5]. Yaks are kept in Mongolia for milk, meat, fiber, hides, and transport. About 80% of the country is covered with grasslands which are suitable for the traditional livestock husbandry practiced in the region, with yak being a major livestock animal that people depend on for both their production and usage in daily life [5]. However, yak production is challenged by factors such as socioeconomic change among herders as a result of privatization, climatic changes (harsh winters combined with droughts), shifts in social values, rangeland degradation, insufficient technologies/infrastructure, poor or no market opportunities, and water scarcity [60,61].

Khalkh Mongols

Khalkh Mongols constitute 81.5% of the total population of Mongolia and represent the core of all Mongolian people across North Asia. Yak keeping forms the prime source of their livelihoods. A transhumant form of yak herding is adopted in Mongolia with seasonal migration [5]. These people celebrate the importance of yak through the yak festival held every 23 July in Orkhon Valley of the Ovokhangai province using the Khangai mountain yaks. This festival is celebrated through various activities such as yak racing, yak showing, wild stallions, yak riding, and milking. The festival starts with sports such as yak racing, continues with yak lassoing, and ends with yak polo [62].

Kazakh

The Kazakh people are a Turkic-speaking minority who constitute the largest non-Mongolian minority in Mongolia with 4.3% of the total population. They form the major population of Mongolia's western aimag, Bayan-Olgiy and some live in Xovd-aimag and dominate one district (sum) called Xovd [63]. Kazakh people in Mongolia are agro-pastoralist, and the size of the herd determines the status of Kazakh people. Yak pastoralism by Kazakh people is based on free-range browsing throughout the year. Some Kazakhs have permanent houses made of wood, stones, and mud in their winter pastures, while most of them live in yurts all year round.

Altai Uriankhains

Altai Uriankhains are located in the northwestern area of Mongol Altai with a population of 1.1% of the national total. Their original language is Tuvinian, but most of them speak Halh as they have adapted to Mongolian culture and tradition. Pastoral nomadism forms an important sector of their livelihood [64]. Yaks are usually grazed in a transhumant rotational migratory system of in higher altitudes where other animals cannot graze [64]. They have a tradition of determining the age of a yak by the pattern of its teeth, and they give nicknames to their yaks based on their color and shape of horns. Altai Uriankhains, being lamaists, do not harm living beings and are extremely hesitant to slaughter animals [64].

*3.7. Nepal*

There are 69,346 head of Yak and nak (yak–cattle hybrids) in Nepal [65], which are reared at an altitude of 3000–5000 m asl. Yak herding is the primary source of livelihood for people residing in the high-altitude Himalayan regions. Sherpa, Tamang, Thakali, Rai, and Limbu are the major ethnic group involved in yak raising in 29 northern districts of Nepal. Yak festivals are organized in Nepal to promote ecotourism, where yak herders from different regions are presented to the tourists and stalls of yak milk products, horse races, yak rides, and local cuisines are kept and folk music and traditional dances are performed. Despite the importance of yak in local livelihood and culture, the yak population has been reducing in recent years due to factors such as endemic diseases, pasture shortage, climatic issues (drought, increased temperature and erratic rainfall, increased extremes), transboundary issues in pasture access, and lack of modern veterinary technologies and facilities [11]. Apart from these, change in social values and modernization has also led to a shift from the traditional yak herding occupation towards lucrative business and tourism jobs in younger generations [66,67]. Another serious challenge for yak herding is the failure of the respective nations and their policies to recognize the importance of high-altitude yak herding cultures on ecosystem regulation and biodiversity conservation [68].

Sherpa

The Sherpas are an ethnic group of Tibetan origin and are mostly settled in surrounding mountain regions like Solukhumbu, Rasuwa, Taplejung, Manang etc. They mainly live in the Solukhumbu regions, located to the south of Mount Everest, and their livelihood are dependent on cultivating crops like potatoes, wheat, barley, and maize and herding cattle and yak–cattle crossbreeds. They adopt transhumant agro-pastoralism, combining both crop cultivation in settled communities and livestock herding away from home to summer and winter pastures over an annual cycle [69]. Yak herding is considered a status symbol among the Sherpa people. Sherpa people from the Solukhumbu district also drink the blood of yak and nak at Lhosar (Buddhist New Year). They believe drinking the blood from these animals will make weak people stronger [10].

Tamang

Ethnically, the Rasuwa district and Limi valley in Humla are dominated by the Tamang people. Among the Tamang community, yak and chauri herding forms a traditional livestock farming practice for those of Tibetan origin, who follow pasture based grazing and transhumant migration in herding [70]. They take their herds towards high alpine pastures in summer and bring them back to lower pastures or forest during winter. In Langtang VDC of Rasuwa district, sale of yaks and yak–cow crossbreeds is a lucrative activity [71]. Likewise, another major economic benefit from the yak in Tamang community is the production of yak cheese. The Tamang people of Rasuwa district also have the tradition of consuming the blood of yak, naks, and both male and female hybrids but, unlike Sherpa people, they do not drink it fresh but rather eat it cooked [10].

Thakali

The Thakali are a Budhhist/Hindu ethnic group of Nepal who speak Tibetan as well as Thakali, a Tibeto–Burman language [10]. The traditional home to agro-pastoral and trader Thakali people [72] is a section of the Thak Khola region of the Upper Kali Gandagi River [73]. Yak production is considered a major source of income by the Thakalis, along with crop cultivation and hotel business. They graze their herds on higher communally owned pastures (3500–4000 m) and the Tibetan plateau (4500 m) during the summer and descend to lower altitudes in winter [72]. Traditionally, fresh yak blood is also drunk by the Thakalis, and it is believed that drinking blood cures digestive problems and helps people with gastritis as well as high blood pressure. Wild yak blood is used to treat fever, dysentery, and schizophrenia, while its bone marrow and tail are used to treat arthritis and snow burn [74]. The blood-drinking ceremony is held twice a year during April/May and July/August in yak pastures, and usually lasts for five days each time [10].

Limbu

The Limbu people are an indigenous group residing in the hilly and mountainous regions between the Arun and Mechi Rivers of east Nepal. They inhabit the Sankhuwasabha, Tehrathum, Dhankuta, Taplejung, Morang, Sunsari, Jhapa, Panchthar, and Ilam districts of Nepal. They call themselves "Yakthumba" in the Limbu language [75]. They keep yak and their hybrids, but primarily keep more female yaks and female hybrids and graze them following transhumant migration system [76].

Rai

The Rai ethnic people are part of one of the oldest ethnolinguistic groups in Nepal. They keep yak primarily in the eastern regions of the Panchthar, Ilam, and Taplejung districts of Nepal and are dependent on yak and yak products as a major part of their livelihood. It has been reported that Rai herders began using the high mountain pastures even before the Sherpas came and later settled in the Sherpa-dominated Khumbu region [77,78].

### 3.8. Pakistan

Northern Pakistan has rich geographical and ethnic diversity, being recognized as one of the most multilingual places in the world. Yak herding in Pakistan is limited to the higher altitudinal areas of northern Pakistan in Gilgit Baltistan and Chitral at altitudes between 3000 to 4000 m asl [16] by ethnic communities like the Baltis, Burusho, Wakhi, Sarakuli, and Yashkun. Yak herding in Pakistan has been affected by disease outbreak, depredation by wild animals, and poor veterinary services [16]. Likewise, changing climatic conditions have caused changes in vegetation composition and reductions in pasture yield in the rangelands of northern Pakistan which has directly impacted yak husbandry [79]. Associated extreme weather events such as droughts and snowstorms have also been reported to cause huge economic losses to the yak herders of Pakistan [79].

### Balti

The Baltis live in the Karakoram mountain range in North Pakistan [80]. They are the descendants of Tibetans and are the only Tibetan group that now follows Islam. It has been reported that this happened as a result of influence through trade, military raids, diplomatic ties, or migration of Muslim peoples to their hometowns. The Balti people follow migratory patterns in livestock husbandry where they take their livestock to high or low pastures depending on the seasons. Yaks are one of the major parts of their livestock herds. All the households in Balti community rear at least one female cow–yak hybrid.

### Burusho

Burusho are the mountain inhabitants of small areas in rugged terrains of the independent Pakistani states Ghizer, Hunza, and Nagar. Although not as popular as in Wakhi community, a few Burusho people also keep yaks in Hunza. They use different words for the yaks kept near their homes for part of the year and for those that are freed year-round in the pastures. The first are called bépay and the latter are called yabá in the Burushaski language.

### Sarakuli

The Sarakuli are settled in the buffer zone of Broghil National Park in Chitral, 285 km away from the district headquarters on the Afghanistan border near the Wakhan Corridor. Altogether 15 families of Sarakuli live in Lashkargaz, with a total population of 44 people and yak herders doing low paid jobs. They speak the Wakhi language. They have 180 yaks and 1000 sheep and goats (source: interview with the head of Sarakuli families, Mr. Umar Rafee in Pakistan).

### Wakhi

The Wakhis live in the northern areas of Pakistan, with major settlements found in the remote and mountainous regions of Chitral and Gilgit-Baltistan [81]. Their major occupation is livestock herding. Wakhi people are agro-pastoralist; thus, they also cultivate wheat or barley. Yak herders of the upper Hunza keep a pure breed of yak but cross-breeding is practiced in Baltistan. Wakhi pastoralists have a herding system quite different than other nomads, depending on a unique mechanism of proper utilization of pasture resources. This system relies on centuries of experience, pastures productivity knowledge, accessibility, water availability in seasons, and vulnerability of herds to predators [82]. Yak herders from the Chitral district also play a special traditional sport related to yak called yak polo [81].

Yashkun

The Yashkun are a tribe residing in Gilgit-Baltistan and in the Kohistan district of Khyber-Pakhtunkhwa, spreading as far as Ladakh and Drass. The Yashkun populations are dominant in Gor, Tangir, Chilas, Darel valley, the Indus Valley below Satin, the upper part of the Gilgit Valley, Punial, Gupis, Astore Valley, and Ghizer [83]. They are believed to have migrated to Northern Pakistan from the Indian subcontinent via the Hindu Kush [84].

*3.9. Russia*

Around 35,000 yaks are kept in somewhat lower altitudes than in China, at about 2500 m to 3500 m, in these mountain areas of Russia [5]. Altainian, Buryat, Tuvan, and Yakut ethnic groups keep yaks in Russia. Herders in Russia use different pastures for their yak herds during summer and winter, as seen in other yak-herding countries, but the use of same pastures all year round in some places has also been reported [5]. Yaks are kept in Russia for milk, meat, and transportation. Lack of winter pastures and rangeland degradation are the major issues for yak-herding ethnic communities in Russia [85].

Altainians

The Altainians live in the Mountain Altai region of the Siberian Altai Republic of Russia. They are a Turkic people speaking Altay and Russian languages and follow Shamanism. Altainians heavily depend on the production from their livestock for their livelihood [86].

Buryats

The Buryat people mainly reside in the Republic of Buryatia in the southern part of eastern Siberia of Russia. They follow Shamanism. Buryats people keep yaks on farms on the Okinsky plateau at altitudes of 1400 to 2500 m on the East Sayan mountains, and 1200–1800 m altitudes in the Zakamensky district of the Lake Baikal watershed [5]. Yaks are left to graze freely during summer in alpine and subalpine pastures, and in lower winter pastures during winter with some supplementary feed. The distinctive feature of their livestock husbandry is the short period and relatively short distance migration [85]. Buryat herders regard all livestock animals as having souls and, therefore, take precautions while slaughtering. They have a tradition of cutting a tuft of hair from the tail of the livestock and sweeping its nostrils while praying for the household's fortune [87].

Tuvans

The Tuvans are an ethnic group native to the West Sayan Mountains in the Republic of Tuva of Russia. Tuvans follow Turkic animistic shamanism alongside Tibetan Buddhism. Traditionally, they have been nomads and live in yurts, moving seasonally from one pasture to another while moving their herds several times a year [88]. Yaks are kept outdoors all year round, with some supplementary feed provided when the ground in Tuva is frozen [5]. They have a tradition of performing rituals where they pray for herd growth and protection [87].

Yakuts

Yakuts people live in the Republic of Sakha in the central part of eastern Siberia of Russia. They are a Turkic ethnic group speaking the Yakut language. Yakuts keep horses, reindeer, and yaks [89]. They started keeping yaks in 1842 after they were introduced from Buryatia. They provide supplementary feed to the yaks even during summer in the initial days of summer grazing, and complete feeding is done in winter because of the harsh climate in this region [5]. They show respect to the livestock while slaughtering and dictate words of apology [87].

*3.10. Tajikistan*

Yak herding in Tajikistan is prevalent in part of the Gorno-Badakhshan district in eastern Pamir. About 14,000 yaks are present in Tajikistan [18]. The ethnic communities that predominantly keep yak are Kyrgyz herders. A few Wakhi people also herd yaks nowadays around stations like Murghab and Langar in Rajon Ishkashim. Kyrgyz and Wakhi yak herders keep yaks in Tajikistan at around 4000 m asl [90]. The transformation process that came with the independence of Tajikistan has had adverse impacts on the economic conditions of the herders due to small herds and insufficient fodder, climatic extremes such as high snowfall resulting in huge death of the yaks (5000 yaks died in 1999), and socioeconomic shifts forcing them to change their traditional occupation [18]. In addition, access to pasture use and poor access to markets is a major challenge which has serious negative implications for herders' livelihoods [91].

Wakhi

The Wakhi people of Ishkashim in the western Pamir of Tajikistan are only non-Kyrgyz yak herders who live in the upper elevations of the Amu Darya valley and in Khargushi Pamir, with control of around 300 yak [18]. Wakhi people in western Pamir follow agro-pastoralism with transhumant movement of herds. All the livestock of the village is taken together in a single group to high pastures ("ailok") by professional shepherds during summer, while in winter, the animals are stall-fed on fodder collected during summer [90]. They move with their family to high pastures, where they stay in a small stone house located in each pasture.

Kyrgyz

Kyrgyz pastoralists mainly inhabit Murghab of eastern Pamir in Tajikistan. The majority of yak herds are still controlled by state-run enterprises or farmers' associations [18]. Murghab is the only region of Tajikistan where pastoralism is practiced all year round. Livestock rearing is the basis of the livelihood of people living outside Murghab town. They follow semi-nomadic pastoralism with various types of complex transhumant movements. Winter fodder is less limited to the Kyrgyz herders in eastern Pamir as there is little snow during winter, although the temperatures are extreme, making winter grazing possible [90]. Yaks are grazed at higher altitudes during winter than other livestock and they are not provided with fodder.

**Table 1.** Ethnic identities of the yak herding communities in the Asian highlands.

| Country | Ethnic Groups | Location | Pastoralism Type | Religion | Use of Yaks |
|---|---|---|---|---|---|
| Afghanistan | Kyrgyz | Big Pamir, Little Pamirs | Nomadic | Sunni Muslims | Milk, meat, dung of yak (kizjak) as their only fuel. They get milk, fur (for clothing and leather), and tail hairs (for brooms) from the yak They barter wool, hides, yak tails, and qurut (cake made out of boiled-down and dehydrated buttermilk) [92]. |
| | Wakhi | Wakhan corridor, Big Pamir, Little Pamir | Semi-nomadic/agro-pastoralist | Ismaili Shia Muslim | Milk, meat, and dung used as fuel and manure, traction power. |
| Bhutan | Brokpas | Sakteng and Merak valleys | Semi-nomadic | Gelugpa Buddhism | Yak products include food, clothing, and housing (tents made of yak hair). Yak dung is used as manure and very rarely as fuel [21]. Skin is processed into jackets and leather bags; tents, ropes, bags, and rugs are made out of coarse outer yak hair (tsipa), and fine inner hair (khullu) is woven into dresses and blankets [29]. |
| | Bjops | Laya, Lunana, Lingzhi, Soe and Nara, Wangdue Phordang | | | The major products from yaks are milk, meat, skin, and hair. Yak skin is used to make floor mats, glue, and kosha (leather meat), and the hairs are turned into tents, ropes, bags, rugs, clothes, and blankets [24]. |
| | La-khaps | Wangdue Phordang, and Trongsa districts | Semi-nomadic | Buddhism | Milk, cheese, and butter. |
| China | Tibetan | Qinghai, Gansu, Yunnan, and Sichuan provinces | Nomadic | Tibetan Buddhism | Milk, milk products, and meat for food. Other products such as hair and hides. Two types of yak hair are used to process into different materials: coarse belly hair is used for tent materials and finer inner wool (khullu) is processed into ropes and blankets [93]. Yak dung is used as a fuel. |
| | Mongolian | Inner Mongolia, Xinjiang, Gansu Province, Qaidam Basin, Kunlun Mountains in Qinghai Province | Nomadic | | Milk and milk products; the distinct milk products from yak are milk skin and milk curd [30]. Hair and meat from livestock are harvested for domestic consumption. Yaks are used as pack animals and for riding |
| | Tajik | Taxkorgan Tajik Autonomous county of Western Xinjiang | | Muslim | Yaks provide milk, meat, wool, and dung for household consumption as well as for bartering. Tajik people trade their livestock products with outsiders in exchange for household items [39]. |
| | Kazakh | Xinjiang Uygur Autonomous Region | Nomadic | | Milk, yogurt, cheese, and meat. |
| | Qiang | Northwestern Sichuan, Tibetan Plateau | Semi-nomadic/agro-pastoralist | Tibetan Buddhism | Milk and milk products, pack animals. Hybrids between yak and cattle are still used for ploughing or transportation in west Sichuan today. |
| | Yugur | Sunan Yugur Autonomous County, Gansu Province | | Tibetan Buddhism | Milk, milk products, meat, and wool |

**Table 1.** *Cont.*

| Country | Ethnic Groups | Location | Pastoralism Type | Religion | Use of Yaks |
|---|---|---|---|---|---|
| India | Bhotia | Tibetan Border in Garhwal, Kumaon region of Northern Uttarakhand | Semi-nomadic/agro-pastoralist | Hinduism | Yak products such as dried meat, wool, and churpi are sold and traded [94]. |
| | Bhuttia | North of Lachung and Lachen Valleys of Sikkim | Semi-nomadic | Buddhism | Milk products including shyow (curd), khachu (whey), marr (butter), thara (butter milk), chhurpi (wet cheese), chilu (yak fat), tema (yak cream), philu (creamy cheese), and shuza/sahpjha or phuicha (butter tea) [50]. They consume a variety of traditionally processed meat products (smoked, sun-dried, air-dried, or fermented), namely satchu (dry meat), kargyong or gyuma (sausages), and chilu (yak fat) [48]. Likewise, various value-added products are made out of yak wool such as ropes, tents, caps, blankets, handbags, door mats, slings, and handwoven carpets. |
| | Changpas | Northern plain of Ladakh and Jammu and Kashmir | Nomadic | Tibetan Buddhism | The milk products from yaks are butter, fermented butter, clarified butter, solidified dried curds, and cheese. The fat and dung of yaks are used as fuel and their meat as food. Wool, hair, and tendons are used to make clothes, tents, carpets, blankets, ropes, and bags. Hide and stomach are used to make storage bags. Yaks are also used to transport the goods of herders when migrating. Stipa or the coarse belly hair is processed into a tent material and soft wool or khullu is woven into ropes and blankets [49]. |
| | Dokpas | Muguthang area of Lhonak valley, Lasher valley, Tsho-Lhamu plateau | Nomadic | | Yak hair is used to make rope and tents, underwool for blankets, skin is used as floor mats, and the tail as a whisk. Different Yak products are consumed and prepared, such as milk, butter, meat (fresh, dry and matures), dry cheese, wet cheese (churrpi), fermented cheese (phyilu), sweetened cheese, cream (tema), and fat (chilu) [50]. |
| | Pangwals | Pangi of Himachal Pradesh | Semi-nomadic | Hinduism small population Buddhism | Milk, milk products, and meat. |
| Kyrgyzstan | Kyrgyz | Issyk- Kul and Naryn Provinces of Eastern Kyrgyzstan | Semi-nomadic | Muslim | Yaks are mainly kept for milk products, meat for consumption, manure, tourist demand, and export [95]. |
| Mongolia | Khalkh Mongols | Western sums of Mongolia in Arkhangay aimag and Khuvsgal | Semi-nomadic | Tibetan Buddhism | Yak milk is used to produce cream, butter, cheese, and yogurt. Milk is also used to produce a special alcoholic drink by fermenting it in a leather pouch and distilling as a "milk wine" (archi) [5]. The coarse outer hair as well as the fine inner hair and hides of yak generate income for the Khalkh people. Likewise, yak meat is also used to produce various meat products such as borts (dried meat). These people use yak for draft purpose to carry household goods, firewood, and hay. |
| | Kazakh | Mongolia Western Aimag, Bayan-Olgiy Xovd-aimag | Semi-nomadic/agro-pastoralist | Buddhism | Yaks are a source of milk and milk products, meat, and wool. |
| | Altai Uriankhains | North-western area of Mongol Altai | Nomadic | Tibetan Buddhism and Shamanism | Yaks are kept for meat and milk as well as for the transportation of goods [64]. Apart from milk, wool and hides of yaks are also extensively used and yak meat is rarely consumed by the Altai Uriankhains. Yaks are used as draft animals. |

**Table 1.** *Cont.*

| Country | Ethnic Groups | Location | Pastoralism Type | Religion | Use of Yaks |
|---------|---------------|----------|------------------|----------|-------------|
| Nepal | Sherpa | Solukhumbu, Rasuwa, Taplejung, and Manang districts | Semi-nomadic/agro-pastoralist | Buddhism | Yaks provide wool and milk and milk products such as butter and cheese. Sherpas use the hybrids of domestic cattle and the yak for carrying trekkers' equipment as many of the Sherpa people are shifting towards the tourism sector from their traditional agriculture and transhumant lifestyle [66]. At the Nara festival in Sherpa villages, butter is used for constructing torma (ritual statuary), for butter lamps, and for frying the special braided Nara bread [69] |
| | Tamang | Rasuwa and Limi valley of the Humla district | Semi-nomadic | Buddhism | Milk and milk products, meat, wools, transportation |
| | Thakali | Thak Khola region of Upper Kali Gandaki River | Semi-nomadic/agro-pastoralist | Buddhism/Hinduism | Milk and milk products such as ghee, butter, cheese. The wool from yak's hair is also used to make ropes, carpets and tents for a shepherd. yak tail is used to scare ill spirits |
| | Limbu | Sankhuwasabha, Tehrathum, and Dhankuta districts | Semi-nomadic | Buddhism/Hinduism | Milk and milk products. |
| | Rai | Panchthar, Ilam, and Taplejung districts | Semi-nomadic | Buddhism/Hinduism | Milk and milk products. |
| Pakistan | Baltis | Karakoram mountain range in North Pakistan | Semi-nomadic | Islam | Milk, butter, qurut, meat, rugs, carpets, socks, gloves, robes, tails used as brooms, leather used for coats. |
| | Burusho | Ghizer, Hunza, Nagar | Semi-nomadic | Islam | Milk, butter, qurut, meat, rugs, carpets, socks, gloves, robes, tails used as brooms. |
| | Sarakuli | Buffer zone of Broghil National Park in Chitral | Nomadic | Islam | Milk, butter, qurut, meat, rugs, carpets, socks, gloves, robes, tails used as brooms, transportation as pack animal, leather used for coats. |
| | Wakhi | Gilgit-Baltistan, Chitral | Nomadic | Ismaili Muslims | Yak products include milk and milk products such as butter (rogan) and cheese (gurut); meat (gosht) and rugs (pulos) are also obtained from yak meat and carpets are made of yak hair [81]. |
| | Yashkun | Gilgit-Baltistan, Kohistan district | Semi-nomadic | Islam | Milk, butter, qurut, meat, rugs, tails used as brooms. |
| Russia | Altainians | Mountain Altai region of Siberian Altai Republic | Nomadic | Shamanism | Yak milk is used to make cheese, which is bartered as well as consumed at home [4]. |
| | Tuvans | West Sayan Mountains in the Republic of Tuva | Nomadic | Shamanism, Tibetan Buddhism | They consume and sell yak meat and milk products. Some fringe products such as horns, hooves, bone-meal, and yak skins are also bartered [4]. |
| | Buryats | Republic of Buryatia in the southern part of eastern Siberia | Semi-nomadic | Buddhism/Shamanism | Yaks are kept primarily for meat and yaks are not normally milked. |
| | Yakuts | Republic of Sakha, central part of eastern Siberia | Nomadic | Shamanism | Yak milk is used to make cheese, which is bartered as well as consumed at home [4]. |
| Tajikistan | Wakhi | IshkashimAmu Darya valley, Khargushi | Semi-nomadic/agro-pastoralist | Islam | Milk, milk products, meat, and wool. |
| | Kyrgyz | Murghab, eastern Pamir | Semi-nomadic | Islam | Milk, milk products, meat. |

## 4. Discussion

In the early days of humanity, the expansion of the human population into the Asian highlands may have relied heavily on the domestication of yaks; without the yak's capacity to live in these highlands, human civilization might not have established and flourished in these remote areas [5,96]. Thus, the yak is not only a key species for the maintenance of pastoral ecosystem functions, but is also a major element in the pastoralism culture of the Asian highlands. Nowadays, yak husbandry represents the primary source of livelihood for major communities residing in the higher altitudes where other forms of subsistence are rare, although some groups practice agro-pastoralism [5]. In this review, we have built up knowledge base on yak-herding ethnic groups, their associated cultures, and challenges faced by yak-herding communities in the Asian highland countries. Yaks are found distributed in alpine and subalpine regions of the Asian highlands at altitudes ranging from 2000–5000 m in 10 countries, namely Afghanistan, Bhutan, China, India, Kyrgyzstan, Mongolia, Nepal, Pakistan, Russia, and Tajikistan [5,14]. We documented 31 major ethnic groups involved in yak herding in 10 Asian highland countries. Many of these ethnic groups use yaks for multiple purposes in terms of foods and health (milk, milk products, and meat), shelter (yak hair tents), fuel (yak dung), income (wool, hides and animal trade), social status (herd size), and means of transportation in difficult mountain terrain [5,48,81,97]. Many studies have shown that herders livelihoods have improved due to sale of yak products [12,98]. In China, there are well developed yak product processing industries, and most yak herder families escape poverty and achieve prosperity through yak products and live animal trade within and beyond the highlands [30]. In Nepal, the annual income from the sale of yak milk and milk products in the Mustang district amounted to about 72,780 Nepalese Rupees (US $1039), and in the Rasuwa district, the annual income from the sale of milk is almost Rs 121,500 (US $1146) [10,99]. Unfortunately, yak herders are not receiving much benefit from the sale of the products. This is mainly due to lack of modern technologies for the processing and preservation of yak products [12]. Another major challenge is that herders have to struggle to sell their products in markets due to poor and limited access to markets [73]. Due to a lack of proper markets, it has been reported that Kyrgyz pastoralists in Afghan Pamir have to suffer to sell their pastoral products on a regular basis [100]. Therefore, to sustain yak production in future, there is a need to improve the processing and preservation of yak products, establish market linkages, and promote diversification of products so that the market niche for yak products can extend beyond local markets to domestic, regional, and international markets. Many ethnic groups are involved in bartering milk products, wool, and hides from yaks for other necessary household items with lowlands people [18,29,49]. In Bhutan, the Brokpa ethnic group have a unique barter tradition, "drukor", of exchanging yak product for grains, particularly with people from lower altitudes in winter [12,28]. Such a barter system connects the variety of ethnic communities in the landscape of Asian highlands and plays a major role in sustaining food security and nutrition in pastoral and agricultural groups of this region [4,97].

There has always been a mutual relationship between the habitat and cultures of ethnic people. We found that the yak holds key significance in the cultural, religious, and social lives of these people [2,3,8]. The horns and skulls of yak have religious significance, and are often engraved with mantras and placed in religious places [8]. Yak tails are used to make Chamar for deity worship [48]. In many Asian highland countries such as Bhutan, China, Mongolia, and Pakistan, yak festivals are celebrated as an annual event every year, with activities such as yak showing, yak racing, yak riding, yak buzkashi, and yak polo. This type of festival provides a platform for people to connect and extend social relationships. In addition, this type of festival provides opportunities for the promotion of tourism, marketing of local products, and cultural exchange. Such unique ethnic cultures of yak-herding communities are worthy of conservation. Similarly, the blood of the yak is also believed to have medicinal values and is consumed fresh by Sherpas of eastern Nepal during Lhosar, and by the Thakalis of central Nepal during the traditional blood-drinking ceremony (Kateo Kanthuno) [10]. Such traditional culture and knowledge of people involved in yak husbandry contributes to the conservation of biodiversity at high elevations and also provides

important information for conservation science. However, the efficacy of traditional pastoral systems has generally been ignored in natural resources management. Conservation, along with the sustainable use of biodiversity, is possible only when the traditional knowledge and cultures of local communities are integrated with science. There are research gaps in documentation of many ethnic groups' unique cultures and traditions. Thus, proper documentation of traditional knowledge will contribute towards the preservation of culture and indigenous practices used by yak-herding ethnic communities.

Despite the fact that yaks form the main basis of livelihood in the Asian highlands, many studies have shown a reduction in yak-herding culture in many Asian highlands regions, mainly due to declines in yak populations [22]. There exist many environmental, social, and economic challenges to yak-herding culture. As yak are very well adapted to high-altitude, cold environments [101–103], climate change may have negative impacts on yak distribution because of their lack of tolerance to heat [15]. Recent Hindu Kush Himalayan Assessment Report (HIMAP) findings are also of concern, as future estimates have concluded that the HKH region, one of the dominant yak-herding regions, will experience at least 0.3 °C higher temperatures [104]. Similarly, increases in temperature have been reported to have negative impacts on the distribution of yaks, decline in yak populations, pasture degradation at high altitudes, shortages of fodder and feed, and food security, and ultimately to have negative impacts on the livelihoods of yak-herding communities [79,97]. One study in northern Bhutan [26] looked at the consistency of herders' perceptions of climate change, where warming is perceived to cause difficulties in herding and transhumant migration and decreases in productivity, ultimately affecting the herders' livelihoods. Similarly, Tibetan yak herders have reported ecological and environmental changes in northwest Yunnan [31,33]. However, the specific impacts of climate change on yaks and its implications for yak-herding communities are currently unknown and require further research.

Social and economic constraints are also contributing to the reduction in yak-herding culture. Evidences from many studies has shown that the tradition of yak herding is significantly under threat and has become a less attractive occupation due to new market trends other than for yak products, changing social values and awareness among herders about education, increased employment opportunities, and outmigration [68]. Likewise, there is increased modernization in the lives of herders and yak herding, due to its nature of hardship, has become a less attractive occupation for local people [105]. In Bhutan, legalization of Cordyceps collection has resulted in some positive changes, but more undesirable changes in overall yak farming [22]. Yak herders are scattered across the Asian highlands, which makes it difficult to implement a specific development program to support them; thus, development interventions are limited in such areas. Yaks still continue to provide key livelihood options for the majority of mountain inhabitants, and play an equally key role in the conservation and regulation of mountain biodiversity [33]. Challenges faced by the pastoral communities need to be understood and addressed in order to prevent rapid loss of rangeland biodiversity and to conserve traditional knowledge related to pastoralism.

Despite these challenges, yak herding is still prevalent and is a dominant occupation of people living in high altitudes, as it brings immense opportunities to the local community. Yaks are the most suitable livestock to be kept in harsh climatic and physical conditions, with large profits able to be acquired from their products and usage [68,97,101,102]. The indigenous knowledge of local people is still alive within the yak-herding cultures, and there is an increased trend of support from governments, different NGOs, and INGOs for the preservation of yak-herding cultures [11]. There are many opportunities to establish niche markets for some yak products, such as promotion of the functional components of yak milk and meat [97], horn, bone, and hair in local, national, and international markets which will provide added value and economic return to yak herders. At present, yak herders' networks have been formed at national and regional levels, and these networks can be linked with herders in other regions through the World Yak Herders' Association (WHYA) network, which was established to contribute to broader regional and global discourse through the Pastoralist Knowledge Hub initiated by the United Nations Food and Agriculture Organization (FAO). Such networks can

provide platforms to work collectively for the sustainable development of yak farming. In view of this opportunity, there is an urgent need to conserve the traditional cultures and pastoral systems of indigenous yak-herding communities, and thus to conserve the mountain ecosystem.

## 5. Conclusions

The present review focused on the ethnic groups involved in yak herding and their associated cultures, and highlighted the challenges faced by yak-herding communities in 10 Asian highlands countries. Altogether, 31 major ethnic groups were identified in 10 different countries of the Asian highlands. Yak husbandry was found to be an indispensable part of mountain livelihood, equally linked with the social and cultural aspects of the people. Yaks have provided major income opportunities to many indigenous people living across higher altitude regions of Asian countries, who otherwise have very few economic opportunities due to their harsh physical and climatic environments. Apart from livelihood, yaks also hold significant importance in the cultural and religious aspects of indigenous cultures, who celebrate yaks through yak festivals. Furthermore, yak herding and the associated cultures of people contribute to the ecosystem regulation and biodiversity management of fragile high altitudinal mountain environments. Unfortunately, yak rearing and related traditions are losing their charm in the recent era due to the effects of modernization, such as exposure to a new culture and changing social norms and several other environmental issues such as climate change. However, we found gaps in the documentation of many ethnic groups' unique cultures and traditions and the challenges faced by many yak-herding communities. Thus, there is a need for proper documentation of yak-herding ethnic culture and urgent action to be taken to minimize these natural and human challenges and to conserve the traditional pastoral systems and cultures of these ethnic communities. Various innovative multidisciplinary approaches and actions are crucial to minimize challenges and maximize opportunities to promote yak herding as an attractive income opportunity.

1. Firstly, to promote sustainable yak rearing in the region, it will be important to scientifically recognize and identify the impacts of climate change on yaks and their habitats. Future research work and interventions should also focus on nutritional feed, establishing yak breeding centers, improving animal healthcare and veterinary services, initiating free or highly subsidized yak breed improvement schemes, developing a grazing management plan, and introducing high-yielding fodder species in rangeland.
2. Secondly, interventions should focus on adding value to yak products through promotion and establishment of good market mechanisms, and on establishing rural urban connectivity of yak production areas to markets and service centers.

**Author Contributions:** Conceptualization, N.W., R.L., S.J., N.B., Supervision, R.L., N.W., Map preparation—G.D., L.S., Literature review, S.J., L.S., Writing—original draft—S.J., L.S., Writing—review and editing, S.J., L.S., R.L., N.W., N.B., M.I., T.D. All authors have read and agreed to the published version of the manuscript.

**Funding:** We would like to thank the German Federal Ministry of Economic Cooperation and Development and the German International Cooperation (GIZ), Swedish International Development Cooperation Agency (SIDA) and Austrian Development Agency (ADA) for providing financial support.

**Acknowledgments:** The study was carried out under the Hindu Kush Karakoram Pamir Landscape Conservation and Development Initiative (HKPLCDI) of the International Centre for Integrated Mountain Development (ICIMOD). We would like to thank Santiago J. Carralero Benítez for providing the report of World Yak Herders Association and Choduraa Dorzhu for providing information on ethnic group from Russia. This study was partially supported by core funds from ICIMOD contributed by the Governments of Afghanistan, Australia, Austria, Bangladesh, Bhutan, China, India, Myanmar, Nepal, Norway, Pakistan, Sweden, and Switzerland.

**Conflicts of Interest:** The authors declare no conflict of interest.

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
