# Peer review of "Ethnic and Cultural Diversity amongst Yak Herding Communities in the Asian Highlands"

_sustainability, doi:10.3390/su12030957_

Round 1

Reviewer 1 Report

The paper has a very descriptive approach. From a geographical perspective, I believe the links between pastoralism, culture and land use are very interesting, especially with respect to the changes imposed by climate change.

Author Response

Thank you for the constructive feedback and suggestion. We have substantially revised this manuscript, moderating our interpretations. We have strictly followed the guideline of the sustainability journal and have modified the manuscript accordingly. We have added the Table and modified the figure in this revised version.

Reviewer 2 Report

The paper deals with a relevant topic, the challenges facing extensive yak farming among the herding communities in the Asian highlands in a scenario of climate change and increased social modernization. In addition, the article is very informative and provides a large amount of data to understand the dimensions of the phenomenon. In this sense, it is a very interesting text to satisfy intellectual curiosity on this subject.

However, the text presents a fundamental problem: it has no structure nor form of a research article. It is rather a purely descriptive text that collects information from a wide literature review, but without adding any remarkable contribution to the already existing knowledge. The result is a kind of Wikipedia of the communities shepherding the yak in the mountains of Central Asia.

In any case, it must be recognized that the authors have compiled a series of potentially very relevant data, but more than just describing them, they should have tried to elaborate some hypothesis to be tested in relation to the current debates on the subject. For instance, on the debate about the role of extensive livestock in a scenario of climate change and energy crisis. Or maybe around the social, economic or cultural transformations that have occurred over time in the management and care of the yak and its consequences on sustainability of the sector. Or maybe other related debates that in some time are mentioned in the text but not developed.

My suggestion is that the authors choose a more ambitious objective than the mere description of the communities shepherding yaks, trying to contribute to the current debates on grazing sustainability. For this, some explicit hypotheses and a methodology to contrast them should be considered and defined (something missing in the current text).

Data about each community should be more homogeneous than those currently presented. Now, in some cases the inhabitants are mentioned, the number of head of cattle and the uses that are given to livestock products, while in other cases these variables are ignored and others such as religious rituals or archaeological evidences are mentioned. For a research article it would be more useful to establish a series of comparative variables that could be observed in various types of communities.

Another way of transforming the current text into a research article could be analyzing the available and the missing data of each community, trying to draw conclusions from it. It would be interesting to have a comparative chart or table highlighting the available information on each community and the missing data, proposing a research agenda for filling the gaps. Besides, a comparative table would allow to develop possible typologies, helping in this way to better understand the relationship between herding yaks and sustainability.

Section 4 is where the main challenges and opportunities of yak herding are synthesized. However, this synthesis does not come from the analysis of the previous data but from the literature review, which is partial and disconnected from the extensive section 3. This section 4 deserves to be expanded providing clearer evidences that illustrates those challenges and opportunities.

Section 5 (Discussion) is very similar to 4, when in fact it should have served to indicate the contributions of the article to the literature debates. Something that is not done. Besides, unjustified statements are made such as, for example, “there is an urgent need for developing policies and programs to conserve the traditional cultures and pastoral system of indigenous Yak herding communities, consequently to conserve the mountain ecosystem” (lines 756-758). Although it may be a reasonable conclusion, it is not clear how it has been reached. As it stands now it seems rather a subjective opinion of the authors. They should better base their argument to reach it.

The same happens in section 6 (conclusions), where it ends by proposing a series of measures that would require a more solid foundation. In fact, some of them seem inconsistent with some previous arguments. For example, when it is proposed to introduce scientifically designed development programs or market mechanisms to boost the Yak's economy, it is not clear to what extent it would be consistent with the proposals to maintain the cultures and livelihoods of local communities (at least, what has happened in the rest of the world for several decades has been the opposite).

In short, although it is a relevant issue and the authors are providing very interesting data, the current paper looks more as an informative book chapter than a research paper. I would suggest writing it again and transforming it into a research article.

Author Response

Thank you for your constructive feedback and suggestion. We have substantially improved this manuscript moderating the interpretation. Please see the attachment for the response to reviewers.

Round 2

Reviewer 2 Report

The text is much better structured than in the previous version and has resolved several of the deficits noted. However, it still has a format that is too far from what a research article should be. There are no hypotheses to prove nor contributions to any theoretical debate on the field. That is, the paper remains eminently descriptive, although it may constitute a good foundation for potential future research on this subject, since it offers a relatively well systematized set of information. Surely the approach of the article does not allow the authors to go further.